# Safety and Effectiveness of an Exercise-Based Telerehabilitation Program in Myalgic Encephalomyelitis and Post COVID Syndrome: Protocol for a Randomized Controlled Clinical Trial

**DOI:** 10.3390/healthcare13233062

**Published:** 2025-11-26

**Authors:** Hermann Fricke-Comellas, Marta Infante-Cano, Alberto Marcos Heredia-Rizo, Ariadna Martín-Fernández, Pablo Escudero-Pérez, Lourdes María Fernández-Seguín

**Affiliations:** 1Instituto de Biomedicina de Sevilla—IBiS, Hospitales Universitarios Virgen del Rocío y Macarena/Consejo Superior de Investigaciones Científicas (CSIC)/Universidad de Sevilla, 41013 Sevilla, Spain; 2CTS 1110: Understanding Movement and Self in Health from Science (UMSS) Research Group, 41009 Andalusia, Spain; 3Departamento de Fisioterapia, Facultad de Enfermería, Fisioterapia y Podología, Universidad de Sevilla, 41009 Seville, Spain; 4Private Practice, Sannu Fisioterapia, 41003 Seville, Spain

**Keywords:** exercise movement techniques, Myalgic Encephalomyelitis, Chronic Fatigue Syndrome, Post COVID Syndrome, telemedicine, self-management, pacing, Mind–body therapies, exercise

## Abstract

**Background/Objectives:** Myalgic Encephalomyelitis/Chronic Fatigue Syndrome (ME/CFS) and Post-COVID Syndrome (PCS) are chronic conditions that share relevant pathophysiological mechanisms. Conventional rehabilitation programs have often been associated with patient dissatisfaction and frequent adverse events (AEs), highlighting the need for safer and more effective clinical approaches. This study aims to compare the effects of a telerehabilitation program based on conscious movement with those of conventional low-intensity exercise in individuals with ME/CFS or PCS. **Methods:** This is a prospective, single-blind, three-arm, parallel, superiority randomized clinical trial. A total of 147 participants (aged 18–70) with ME/CFS or PCS will be recruited and randomly assigned to one of three groups: (a) conscious movement; (b) low-intensity exercise; or c) usual care. All interventions will be delivered via telehealth over 12 weeks, with weekly 45-min sessions combining health education and individually tailored exercises. Participants will be encouraged to practice daily using the provided materials. Adherence rates and potential AEs will be recorded. The primary outcome is the total score on the 14-item Chalder Fatigue Scale at 12 weeks (post-intervention). Secondary outcomes include heart rate variability, functional performance, pain intensity and interference, mental health, interoceptive awareness, quality of life, sleep quality and fear of movement. Measurements will be collected at baseline, post-intervention, and at 3-month follow-up. **Discussion:** Recent evidence suggests that both autonomic and cognitive activity modulate immune function. Conscious movement, which integrates exercise with interoception and mindfulness-based strategies, may provide greater benefits than low-intensity exercise alone. Study limitations should be considered when interpreting the results. **Trial registration:** Registered at ClinicalTrials.gov on 15 May 2025 (NCT06978582). Protocol version 4 (29 September 2025). Ethics Committee code: 2025-0180.

## 1. Introduction

Myalgic Encephalomyelitis/Chronic Fatigue Syndrome (ME/CFS) is a chronic condition characterized by severe, medically unexplained fatigue [1], which imposes major socioeconomic costs by limiting daily function and work capacity [2]. Patients frequently report feeling misunderstood by healthcare providers, who often underestimate the impact of fatigue [3]. Additional symptoms include post-exertional malaise (PEM), dyspnea, pain and sensory hypersensitivity, vasomotor, bladder, and gastrointestinal dysfunction, cognitive and sleep problems, and muscle weakness—all of which is consistent with dysautonomia [1]. Post-COVID syndrome (PCS) refers to a cluster of symptoms that persist or emerge after the acute phase of SARS-CoV-2 infection [4]. Overlapping symptoms and shared pathophysiological mechanisms, particularly autonomic dysfunction, have become increasingly evident between ME/CFS and PCS. Several authors suggest the importance of cross-fertilization of research across these conditions, including the use of mixed-cohort trials [5,6,7]. A central aspect of ME/CFS and PCS is interoception, defined as the perception of internal bodily states. Interoceptive signals underlie autonomic reflexes, motivational drives, and conscious sensations that guide adaptative behavior [8]. Disruption of interoceptive processing may therefore play a critical role in the clinical presentation of ME/CFS and PCS [9,10].

Currently, there are no disease-specific pharmacological treatments for ME/CFS [11] or PCS [12]. Non-pharmacological approaches aim to reduce fatigue and improve functional capacity through rehabilitation, anti-inflammatory measures, and dietary interventions. Low-intensity exercise with strict pacing is recommended for individuals diagnosed with ME/CFS who are improving and wish to increase activity levels [13]. Such intervention may help relieve symptoms and prevent deconditioning in ME/CFS [11] and PCS [12]. However, PEM is common and greatly narrows the therapeutic window [14]. PEM involves a worsening of symptoms after minimal physical, cognitive, emotional, or social activity, with exacerbations typically occurring 12 to 48 h later and lasting for days or weeks. Potential adverse events (AEs) therefore include PEM episodes and symptom flares, such as increased fatigue, cognitive difficulties, pain, sleep disturbance, and orthostatic intolerance. This requires careful dose titration, planned recovery, and continuous patient feedback [13]. 

Previous exercise trials in ME/CFS have reported AEs inconsistently and have often lacked patient-centered dosing approaches, undermining confidence in exercise-based rehabilitation [15,16,17]. Current clinical guidelines advise against generic graded exercise prescriptions and recommend caution, especially in severe disease [13]. In PCS, structured, remotely delivered programs, alone or combined with education, have improved quality of life and related outcomes [18,19,20,21,22,23,24,25]. Yet, reporting of AEs remains inconsistent [26]. When therapeutic programs are individualized, symptom-titrated, and closely monitored, telerehabilitation appears feasible and generally safe [26]. 

Conscious movement-based exercise, or mind–body exercise (MBE), may reduce physical activity-related AEs by fostering pacing and symptom-titrated effort. Its core feature is movement awareness and continuous appraisal of bodily sensations and context [27]. Sustained interoceptive focus may enhance interoceptive awareness in people with chronic conditions [28], enabling real-time adjustments in effort and recovery, thereby reducing the risk of PEM. Evidence suggests that MBE produces similar effects on pain and fatigue to those of low-intensity aerobic training [29], although direct head-to-head comparisons with active exercise controls remain scarce [29,30]. Meta-analyses have shown small-to-moderate reductions in fatigue with MBE in ME/CFS and PCS (SMD −0.44, 95% CI −0.63 to −0.25) [29]. Beyond physical impact, MBE incorporates benefits associated with meditation, mindfulness practices, and conscious breathing. These include analgesia through cognitive reappraisal [31], reductions in perceived stress [32], and modulation of anti-inflammatory and cytokine activity [33], as well as immune regulation and stress-buffering mechanisms [34]. Considering the well-established links between stress, sympathetic overactivation, and Hypothalamus–Pituitary–Adrenal (HPA) axis dysregulation, MBE and related practices may contribute to restoring autonomic balance [32,33]. 

Despite promising findings, key evidence gaps remain for the efficacy of telerehabilitation and MBE. Telerehabilitation is under characterized in ME/CFS, and most existing evidence derives from post-COVID cohorts. Few studies have compared MBE with active exercise interventions. Reporting of safety and AEs is inconsistent, and mechanisms such as interoceptive awareness and autonomic balance are rarely assessed. Similarly, the durability of effects beyond the treatment period is still uncertain. 

To address these gaps, this protocol proposes a three-arm, head-to-head design, with PEM-aware dosing, strict AEs monitoring, inclusion of process measures (interoceptive awareness, heart rate variability—HRV), and follow-up at 3 months. The primary aim is to determine whether an MBE telerehabilitation program reduces fatigue severity, as measured with the 14-item Chalder Fatigue Questionnaire (CFQ-14), more effectively than conventional low-intensity exercise after 12 weeks of intervention. Secondary aims are to: (i) describe safety outcomes, with a focus on AEs under a pacing and recovery strategy; (ii) assess changes in dysautonomia-related symptoms; and (iii) examine the durability of effects 3 months after intervention. We hypothesize that the MBE program will enhance autonomic balance and interoceptive awareness compared with conventional exercise, facilitating more adaptative pacing and recovery behaviors, and therefore reducing the frequency and severity of PEM. Consequently, greater improvements in CFQ-14 scores are expected in the MBE group at 12 weeks. Although low AEs rates are anticipated under careful pacing and monitoring across groups, we hypothesize that the durability of effects will depend on participants’ adherence to the program after completion of the intervention.

## 2. Materials and Methods

### 2.1. Trial Design 

A single-blind, three-arm, parallel-group, superiority randomized controlled trial will be conducted. The protocol adheres to the Standard Protocol Items Recommendations for Interventional Trials (SPIRIT) guideline [35], and incorporates the Template for Intervention Description and Replication (TIDieR) checklist [36], and the Consensus on Exercise Reporting Template (CERT) [37]. Technical patient groups from the Spanish Research Network on Post-COVID (REiCOP) contributed to protocol development, providing input on feasibility, pacing safeguards, and patient-relevant outcomes. The study protocol has been approved by the Institutional Human Research Ethics Committee (code: 2025-0180) and registered at ClinicalTrials.gov (NCT06978582). Figure 1 presents the study protocol flowchart. 

### 2.2. Study Setting

The study is decentralized and fully remote. A single coordinating center will oversee recruitment, randomization, intervention delivery, data management, and safety monitoring. No in-person visits will be required. 

The study team comprises two senior physiotherapists and certified yoga instructors with more than 10 years of experience in therapeutic yoga, one physiotherapist with over 5 years of experience in therapeutic exercise for chronic fatigue and complex pain, three researchers responsible for online assessments, and one statistician in charge of data analysis. To ensure consistency across evaluations and intervention delivery, preparatory training sessions will be conducted prior to recruitment.

### 2.3. Eligibility Criteria

Inclusion criteria are as follows: individuals aged 18–70 years who meet the Institute of Medicine (IOM) diagnostic criteria for ME/CFS [1], or have a history of confirmed COVID-19 and fulfill diagnostic criteria for PCS [4]. Participants must have access to a computer, tablet, or smartphone for videoconferencing and be able to download and use the Welltory mobile app, V. 4.37.0 (Welltory Technologies Inc., Redwood City, CA, USA) to assess HRV [38]. They should be able to maintain a seated position for at least 45 min and demonstrate fluent comprehension of Spanish. Exclusion criteria include suspected but unconfirmed comorbidities, inability to use the software required for the study (e.g., Microsoft Teams, Google Forms), and unwillingness to adopt minor daily habit adjustments.

### 2.4. Diagnosis Confirmation and Disease Duration

A prior formal ME/CFS or PCS diagnosis is not required due to very limited specialist access withing European clinical settings [39,40]. All participants must provide documentation of a prior medical evaluation reasonably excluding alternative diagnoses (e.g., post-infectious organ damage, cardiovascular or neurological disease, cancer). The research team will not perform medical diagnoses. The lead investigator will verify that participants’ medical history meets the diagnostic criteria using a structured checklist and review of available records. Disease duration will be defined as the number of months from the self-reported onset of persistent symptoms meeting diagnostic criteria to the baseline assessment. For PCS, the date of the acute SARS-CoV-2 infection will also be recorded.

### 2.5. Consent or Assent 

The lead researcher will obtain both verbal and written informed consent from all participants. They will be informed of the study aims, potential risks and benefits, and their right to withdraw at any time by notifying the research team. Each participant will be assigned with a unique identification code linked to their personal information, which will be stored securely and accessible only to one designated researcher. The consent form includes a statement specifying that collected data may be used for future research, subject to additional ethical approval. 

### 2.6. Recruitment

Recruitment will be conducted in collaboration with REiCOP and affiliated patient associations. These organizations will disseminate information about the study aims, eligibility criteria, and contact details for the research team. A call for participation will also be posted on social media. Interested individuals will contact the research team to receive an information sheet and schedule a first online meeting with the lead researcher, during which a brief structured interview will verify eligibility and provide detailed information on study procedures.

### 2.7. Sample Size

Sample size estimation considered three study groups, a one-tailed test (a larger effect size is expected in the MBE group), an α value of 0.05, a type II error (β) of 0.20 (80% power), and a precision of 6 (equivalent to a 15% change in the primary outcome, CFQ-14 total score at 12 weeks). Fatigue variance was set at 56.5 based on a previous study with a comparable population, intervention, and measurement tools [41]. Allowing for a 20% dropout rate and a design effect of 2 (due to non-probabilistic convenience sampling), a total of 147 participants (49 per group) will be required. Calculations were made using software developed by the Clinical Epidemiology and Biostatistics Unit of the Complexo Hospitalario Universitario de A Coruña, Spain [42].

### 2.8. Allocation to Interventions

Eligible participants will be randomly assigned to one of three arms: (1) MBE, (2) conventional low-intensity exercise, or (3) usual care. Recruitment will occur in three waves (*n* = 50, 49, and 48), each randomized independently to ensure a 1:1:1 ratio (16–17 participants per arm per wave). Randomization sequences will be computer-generated in R (with seed set for reproducibility) using stratified lists to minimize imbalance across groups. The R script for the stratified randomization is in Appendix A. Allocation concealment will be ensured with sequentially numbered, opaque, sealed envelopes, prepared by an independent collaborator not involved in recruitment or assessment. Envelopes will be opened only after consent and completion of baseline assessments.

### 2.9. Blinding

Outcome assessors and the data analyst will remain blinded to group allocation. Given the nature of the interventions, i.e., exercise and education sessions, blinding participants and therapists is not feasible. A password-protected emergency unmasking system will be established. Unblinding will be authorized only when strictly necessary for safety reasons to preserve trial integrity.

### 2.10. Interventions

Two active, therapist-guided telerehabilitation protocols will be combined with education and compared with usual care: (a) an MBE program, and (b) a low-intensity aerobic and strength training program. Usual care will reflect routine medical management. All sessions will be delivered live Via Microsoft Teams, with groups of ≤7–8 participants, once weekly for 12 consecutive weeks, consistent with previous research in ME/CFS and PCS [29,43]. Each session will last 45 min, increasing by 5 min each month. Exercises will require only a chair, with both seated and standing options available. Adjustments based on physical performance and perceived fatigue or dyspnea will follow pacing principles to minimize PEM risk. Therapists will guide participants using in-session ratings of perceived exertion (2–3/10) and simple cues to maintain a safe and comfortable exercise practice. Participants may also pause or re-join sessions as needed and will be encouraged to practice regularly using recorded materials. Attendance to ≥70% of the 12 scheduled sessions will be considered adequate exposure. A session will be considered attended if the participant joins the live class and takes part at least partially (≥15 min or completion of ≥1 exercise block), in line with pacing allowances. Attendance will be tracked Via Microsoft Teams and verified by therapist logs.

#### 2.10.1. MBE Group

This group will receive a therapeutic program combining health education (15 min) and MBE (30 min). Educational content will include self-management, pacing, activity–rest planning, basic theoretical framework about the conditions, and familiarization with mindfulness and awareness practices and their purported benefits. To enhance proprioceptive and interoceptive awareness, therapists will follow a brief checklist (posture, breath, body-scan, pacing, and cognitive reappraisal cues) at the beginning of each session. Body awareness will be reinforced through short post-session exercises (attention to bodily sensations; effort rating using sensations; perceived calm), and weekly home-practice using prerecorded materials. Specific content is provided in Appendix A. The 30-min MBE practice will consist of structured sequences of breathwork, chair or standing-based mindful movement (yoga inspired), and guided body awareness and relaxation (e.g., Yoga Nidra). Content will rotate across three themes (whole body, upper limbs, and lower limbs) and repeated throughout 12 weeks. Posture lists and progressions follow standardized guidelines [44], and are described in Appendix A. Participants will continue with their usual medical care.

#### 2.10.2. Aerobic/Strength Low-Intensity Exercise Group

This group will receive a program combining health education (15 min) and exercise (30 min). Educational content mirrors that of the MBE group but emphasizes the role of exercise in symptom management. Specific content is provided in Appendix A. The exercise component includes low-intensity strengthening using body weight, gentle aerobic-functional drills, and cool-down and stretching. Typical activities involve joint mobility, multi-joint strengthening with short sets, and simple aerobic drills adapted for home practice, with a calm return-to-rest routine. Program details and progression guidelines are listed in Appendix A. Participants will continue with their usual medical care.

#### 2.10.3. Control Group (Usual Care)

Control participants will continue their usual medical care during the 12-week period.

#### 2.10.4. Criteria for Discontinuing or Modifying Allocated Interventions

Participants may pause or modify sessions at any time according to pacing principles. Therapists may recommend temporary pauses or adjustments based on symptoms. No cross-over between study arms will be permitted. The principal investigator (PI) will decide on discontinuation if safety concerns arise. Withdrawals will be respected without requiring justification. Temporary interruptions due to intercurrent illness or symptom flare will be permitted, with resumption when appropriate. Enrollment in other clinical trials will not be allowed. Any relevant changes in medical treatment will be documented but will not necessarily lead to withdrawal unless recommended by a physician. In the event of AEs, participants will be offered an individual videocall to address causes and adjust the intervention accordingly. More specific details are included in Appendix A.

#### 2.10.5. Strategies to Improve Adherence to Interventions

Adherence will be promoted through structured education, short home-practice videos, a moderated WhatsApp group for reminders and troubleshooting, and individualized autoregulation (pacing, and option to switch exercise version or position). The Welltory app will be used to log daily fatigue, discomfort, and PEM warnings. Entries are time-stamped, and symptom logs are aligned with HRV data for safety reviews.

#### 2.10.6. Provisions for Post-Trial Care

Both programs are low-intensity, self-paced, and adaptable, thus no serious AEs are expected. After study completion, control participants will be offered access to the recorded sessions and materials of either intervention program.

### 2.11. Outcomes

Data will be collected using Microsoft Forms and videoconference assessments. Participants will complete online questionnaires within a 2-week window at their own pace. The PI will monitor submissions for completeness, and plausibility/inconsistency issues. Implausible or inconsistent entries prompt participant queries. Corrections will be documented without overwriting original data. During this 2-week period, participants will attend a ~20-min videocall to assess HRV first and then functional performance in a standardized sequence: Modified Sit-and-Reach Test (MSRT), One-Leg Stand Test (OLST), and 60-s Sit-to-Stand Test (STS-60), to limit fatigue carry-over. Videocalls will be scheduled Monday–Thursday, 10:00–13:00 and 16:00–19:00 avoiding meals and sleep times. Rescheduling will be allowed during flares. This approach minimizes participant burden and reduces the risk that lengthy procedures may influence performance or clinical status.

#### 2.11.1. Primary Outcome

The primary outcome is the total score of the CFQ-14 (Spanish version) at 12 weeks (post-intervention). The CFQ-14 measures perceived physical and mental fatigue over previous weeks on a 4-point Likert scale (0 = “never”, 1 = “sometimes”, 2 = “often”, 3 = “always”), total score 0 to 42, with higher scores denoting greater fatigue. The Spanish version of the CFQ-14 has demonstrated adequate reliability and validity [45]. 

#### 2.11.2. Secondary Outcomes

Secondary outcomes include physiological, psychosocial, and functional performance measures.

Resting heart rate (HR) and HRV will be assessed using the Welltory mobile app (V4.16.100, Welltory Technologies Inc., Redwood City, CA, USA). Resting HR values will be obtained via photoplethysmography through the smartphone camera, while the time-domain HRV indices will be derived as proxies of autonomic balance. Previous studies report excellent agreement to estimate resting HRV using this method when compared with established referenced tools, including electrocardiography and medical-grade HR monitors (intraclass correlation coefficient > 0.90) [46]. Measurements will be made at the start of the video call. Participants will be instructed to sit quietly, breathe spontaneously, rest their forearms on their thighs, and hold the smartphone in their dominant hand, placing the index finger on the camera as indicated by the app. Speaking or moving during the recording will not be permitted. The 300-beat setting will be used to maximize precision. Both resting HR and HRV will be extracted from the same measurement.

Health-related quality of life will be evaluated with the 36-Item Short Form Health Survey (SF-36). This tool comprises eight domain scores (physical functioning, role-physical, bodily pain, general health, vitality, social functioning, role-emotional, mental health), each transformed into a 0–100 scale (higher scores indicate better health). Physical and mental component summary scores will also be calculated. The Spanish version of the SF-36 exhibits high internal consistency across domains [47]. The severity of anxiety and depressive symptoms will be assessed using the Hospital Anxiety and Depression Scale (HADS; Spanish version). The tool includes 14 items forming two 7-item subscales (from 0 to 21 points each). Scores ≥ 8 indicate clinically relevant symptoms. The Spanish version of the HADS is reliable and widely used in clinical and research settings [48]. Sleep quality will be assessed using the Spanish version of the Pittsburgh Sleep Quality Index (PSQI). The PSQI includes 19 items grouped in 7 components, namely, subjective quality, latency, duration, habitual efficiency, disturbances, medication use, daytime dysfunction, each scored from 0 to 3. The global score ranges from 0 to 21, with values >5 indicating clinically significant sleep problems. The Spanish version of the PSQI shows good validity and internal consistency [49].

Remote assessment of functional performance has been shown to be both feasible and reliable when delivered under standardized protocols and real-time supervision [50]. The study will follow international recommendations for remote physical performance testing [50]. Before assessment, participants will receive clear instructions to prepare a well-lit space and position the camera to provide a full-body lateral view. Each test will be demonstrated in advance, and corrective feedback will be provided as needed to ensure proper execution. 

Posterior chain flexibility will be measured using the MSRT. Participants will sit on a firm surface (e.g., floor) with legs extended, ankles neutral, and a one-fist gap between the knees. A small object (≤3–4 cm diameter) will be placed between the legs near the pelvis, aligned with the midline. With arms extended, participants will slide the object forward along the midline as far as possible while keeping knees extended and avoiding bouncing. The assessor will view from the side to ensure correct execution. Using a tape measure, the distance will be recorded in centimeters from the pubic symphysis (over clothing) to the nearest edge of the object at the farthest reach. Testing will stop immediately if pain or cramping occurs. Higher values indicate greater flexibility. Static balance will be assessed using the OLST. Participants will stand unaided on one leg with arms crossed over the chest, while keeping the contralateral leg off the ground. The maximum time (in seconds) that balance is maintained will be recorded, with the best performance of three consecutive attempts used for analysis. Lower-limb strength and endurance will be evaluated with the STS-60. Participants will perform repeated sit-to-stand movements from a standard armless chair for 60 s, with arms crossed over the chest. Effort will be self-regulated to a perceived exertion of approximately 6/10 (moderate). Participants will be instructed to maintain a comfortable pace and may slow down or pause if symptoms increase. The total number of valid repetitions (full stand and controlled sit) will be recorded.

#### 2.11.3. Other Secondary Measures

The Brief Pain Inventory (BPI; Spanish version) will be used to record pain intensity and interference. The BPI comprises 9 items addressing pain intensity and the degree to which pain interferes with daily activities such as sleep, mood, and work. Items are rated from 0 to 10, with higher scores indicating greater pain or interference. The Spanish version of the BPI demonstrates robust validity and reliability [51]. Interoceptive awareness and changes in body awareness will be measured using the Multidimensional Assessment of Interoceptive Awareness (MAIA) scale. This tool includes 23 items rated from 0 (never) to 5 (always) across eight domains (e.g., noticing, emotional regulation Via bodily sensations, interoceptive trust). Higher scores reflect greater interoceptive awareness [52]. Finally, pain-related fear and somatic vigilance will be evaluated using the Tampa Scale for Kinesiophobia (TSK; Spanish version) and the Anxiety Sensitivity Index (ASI; Spanish version). The TSK includes 17 items rated from 1 to 4, with higher scores indicating greater fear of movement [53]. The ASI consists of 16 items (total score: 0–64) assessing fear of anxiety-related bodily sensations, with higher scores denoting greater sensitivity [54].

#### 2.11.4. Adherence and Adverse Events

Adherence and AEs will be monitored using a digital daily diary (Microsoft Forms). Participants will record their completion of prescribed practice sessions, perceived exercise-related sensations, and any AEs or post-exertional symptom exacerbations. The diary will also capture health-relevant events, including menstruation (for women of reproductive age), intercurrent infections or reinfections, and recent vaccinations. A short free-text field will allow participants to note other relevant information (e.g., medication changes, acute stressors). All entries will be used to inform safety reviews and sensitivity analyses.

### 2.12. Data Collection, Retention, and Management

Only health-related data relevant to the study will be collected. For sociodemographic, clinical, and questionnaire-based data collection, links to Microsoft Forms will be sent to participants on the scheduled day, and an assessor will be available to address questions. Several engagement strategies will be implemented to maximize retention and reduce dropout, including providing clear and comprehensive information about the trial, maintaining direct communication with the research team through WhatsApp groups to foster trust and feedback, and sending regular reminders of upcoming sessions, diary completion, and follow-up assessments.

Personal information will be pseudonymized using unique alphanumeric codes for each participant. The file linking codes to participant identities will be securely stored in an encrypted, password-protected document on the institutional cloud server (Microsoft OneDrive), with access restricted to authorized data managers. All study data will remain within the institutional secure cloud environment, ensuring confidentiality, compliance with data protection policies, and safe collaborative work among the research team.

### 2.13. Participant Timeline

A schematic diagram of the study timeline is presented in Figure 2 and Figure 3.

### 2.14. Statistical Methods

Statistical analyses will be conducted using IBM SPSS Statistics, version 26.0 (IBM Corp., Armonk, NY, USA), under an intention-to-treat framework. A complementary per-protocol analysis will include participants achieving ≥ 70% adherence. Baseline characteristics will be summarized as mean (standard deviation) or median (interquartile range, Q1–Q3) for continuous variables, and as frequencies (%) for categorical variables. Between-group comparability at baseline will be examined using chi-square or Fisher’s exact tests for categorical variables, and one-way analysis of variance (ANOVA) or Kruskal–Wallis tests for continuous variables, according to distributional assumptions. Normality will be assessed with the Shapiro–Wilk test and by visual inspection of histograms and Q–Q plots to identify skewness, kurtosis, or outliers. When appropriate, homogeneity of variances will be verified with Levene’s test.

The primary outcome (CFQ-14 total score at 12 weeks, post-intervention) will be analyzed using linear mixed-effects models (LMMs) with fixed effects for group, time, and their interaction (group × time), and random intercepts to account for within-subject correlation. This approach accommodates repeated measures and missing data under a missing-at-random assumption [55]. Estimated marginal means with 95% confidence intervals (CIs) will be reported. Effect sizes will be presented as standardized mean differences (Cohen’s d) and partial eta squared (η^2^).

Secondary continuous outcomes (e.g., HRV, functional performance, pain, anxiety and depression, quality of life) will be analyzed using the same LMM framework. Model assumptions will be verified through residual diagnostics; if violations are detected, data transformations will be considered, or generalized LMMs with appropriate distributions (e.g., Poisson, gamma, binomial) will be applied. For categorical outcomes, between-group differences will be tested with chi-square or Fisher’s exact tests. Where adjustment for covariates is warranted (e.g., age, sex, baseline fatigue severity), logistic regression analysis will be performed, and odds ratios with their 95% CIs will be reported. No multiplicity adjustment will be applied to secondary outcomes, following common clinical trial reporting standards [56]. Pairwise post-hoc comparisons between groups and across time points will use Bonferroni correction for multiple testing. Unless otherwise specified, all tests will be two-tailed, with statistical significance set at *p* < 0.05, and results will include effect sizes and 95% CIs.

#### Additional Analyses

Exploratory analyses will examine potential moderators and predictors of treatment response (e.g., age, sex, baseline fatigue severity, autonomic function). Linear or logistic regression models will be employed, including interaction terms where appropriate. These analyses are hypothesis-generating; results will be interpreted cautiously and without correction for multiplicity, and effect sizes with 95% CIs will be provided. If substantial missing data is observed, multiple imputation will be undertaken as a sensitivity analysis. Consistent with the intention-to-treat principle, all randomized participants will be included in primary analyses; while the per-protocol analysis will be restricted to those attending ≥ 70% of sessions. Adherence rates may be incorporated as covariates in exploratory models.

### 2.15. Oversight and Monitoring

The PI will oversee trial supervision, coordinate all project activities, monitor progress, and implement corrective actions as needed. The research team will serve as the coordinating center, ensuring adherence to the protocol and maintaining high-quality trial conduct. As no serious AEs are anticipated, no independent data monitoring committee will be established. The PI will be responsible for ensuring trial safety and integrity, and will fully cooperate with any auditing procedures, providing all requested documentation. In the event of a major AE, the PI will promptly contact the participant’s treating physician and the Institutional Ethics Committee to determine the appropriate course of action.

Although no major protocol modifications are expected, any amendments will be first submitted to and approved by the Institutional Ethics Committee before implementation. Participants will be informed of any changes affecting their involvement, and updated study materials (e.g., information sheets) will be provided when necessary.

### 2.16. Dissemination Plans

Study findings will be communicated to participants at the end of the trial. Results will also be disseminated to the scientific and clinical community through presentations at national and international conferences and publications in high-impact, open access journals. Reporting will follow transparent research practices.

The full study protocol and data collection materials will be made publicly available to promote transparency and reproducibility. De-identified datasets and analysis code will be deposited in the institutional open-access repository (https://idus.us.es/home, accessed on 12 October 2025) under a persistent identifier and will remain accessible without time restriction. Source data containing personal information will be securely stored on university servers for 3 years after study close-out and will then be deleted or irreversibly anonymized.

## 3. Discussion

Although the exact pathophysiological mechanisms underlying both ME/CFS and PCS remain unclear, current evidence suggests a central role of the immune system. Persistent systemic inflammation and immune dysregulation may contribute to dysfunction across multiple systems [57,58,59]. Increasing research highlights complex interactions between cognition, the nervous system, and immune function [60,61,62]. Stress and cognitive-emotional states activate the HPA axis and the autonomic nervous system, altering glucocorticoid release and sympathetic–parasympathetic balance, which in turn modulates cytokine production and immune cell activity [63]. In addition, learning and conditioning processes can modify immune responses, underscoring the bidirectional links among brain, behavior, and immunity [64,65]. These findings indicate that therapeutic approaches targeting not only physical but also cognitive, autonomic, and interoceptive domains may achieve broader benefits. In this context, interventions designed to enhance bodily awareness, regulate autonomic activity, and reduce maladaptive cognitive patterns, such as MBE, may complement conventional rehabilitation and promote sustained improvements.

The present trial protocol operationalizes this rationale through a structured telerehabilitation exercise program, offering a feasible and accessible alternative to traditional in-person interventions. By eliminating barriers related to transportation, fatigue, and travel demands, the proposed MBE program is expected to facilitate adherence and reduce the economic burden for both patients and healthcare systems. The telerehabilitation format aligns with current trends in digital health, representing a scalable strategy for implementation in public healthcare settings. In Spain, as in many other countries, adults diagnosed with ME/CFS or PCS often face substantial barriers to adequate care, beginning with difficulties obtaining an accurate diagnosis under accepted clinical criteria, and are compounded by the lack of specialized, coordinated healthcare units. All this leaves patients to navigate fragmented services with limited professional guidance and support [39,40,66], which further exacerbates the burden of living with limited energy and highly disabling symptoms. A structured telerehabilitation program that combines exercise, self-management strategies, and education, with a consistent follow-up from a coordinated multidisciplinary team, could therefore be highly meaningful from a clinical perspective. Even modest improvements in functional capacity and symptom management could translate into substantial changes in quality of life, while addressing the unmet needs of an often-unserved population. In line with emerging research directions [67,68], this study proposes a symptom-titrated telerehabilitation approach that dynamically adjusts exercise load through clear, structured, and comprehensive pacing strategies tailored to individual’s clinical response. The MBE program has been designed to address key limitations of previous telerehabilitation programs and provide a truly personalized care model for patients with ME/CFS and PCS. Therefore, the trial findings may help shape future clinical guidelines and inform public health strategies into standard management of CFS/ME and PCS

Nevertheless, this approach presents inherent limitations. In line with other telemedicine trials, participation requires stable internet access and basic digital literacy, which may restrict the generalizability of findings [69]. Moreover, individuals with cognitive impairments may face challenges engaging with online content, scheduling videoconferences, or using digital applications [50]. These issues may be further complicated when digital tools such as the Welltory app are not available in the participant’s native language. Online assessments may also limit the precision of certain outcome measures compared with face-to-face evaluations. Although all instruments in this trial have been validated, some may not achieve the same accuracy as gold-standard methods available in onsite settings. Biological sampling is also precluded in an online clinical trial. Self-report outcomes are susceptible to expectation and recall bias; we aim to mitigate through triangulation with objective and observer-rated measures. The absence of in-person group dynamics may further reduce potential psychosocial benefits derived from peer interaction and shared experiences, which are commonly observed in traditional rehabilitation environments [70]. These limitations should be carefully considered when interpreting trial results and planning future iterations of the interventions. 

Despite these limitations, this randomized controlled trial will represent a significant advance in three key areas. First, rigorous monitoring of AEs and systematic implementation of pacing strategies are essential to minimize the risk of PEM. Future studies should continue prioritizing safety to ensure that rehabilitation protocols for ME/CFS and PCS are both effective and acceptable for patients [13,71]. Second, this trial will evaluate the impact of integrating an MBE approach within a telerehabilitation format, with a direct comparison with a conventional exercise program commonly used in this population. Third, it addresses a critical gap in the management of ME/CFS and PCS by testing a scalable, low-cost, safe, and patient-centered intervention. If the intervention proves clinically effective, the findings could inform health policy and support the integration of supervised telerehabilitation into standard care pathways for adults with ME/CFS or PCS.

## 4. Conclusions

This trial aims to examine the effects of a 12-week mindful exercise telerehabilitation program compared to conventional exercise and usual care in individuals with ME/CFS and PCS. Beyond testing feasibility and clinical outcomes, this study seeks to elucidate whether an interoceptive, symptom-titrated approach can support safer and more sustainable physical activity engagement.

Findings from this trial may help refine personalized rehabilitation strategies and inform the integration of telerehabilitation into routine clinical pathways for ME/CFS and PCS. By providing a structured, evidence-based model of remote delivery, the protocol could serve as a foundation for future multicenter or cross-national collaborations and guide health policy decisions toward broader access to individualized, home-based rehabilitation options.

## 5. Trial Status

The trial is ongoing and is currently during the recruitment phase. Recruitment was initiated on 1 July 2025, and is expected to be completed by the end of January 2026. 

Protocol version 4 (29 September 2025). The trial was registered at ClinicalTrials.gov (NCT06978582) on 15 May 2025, and the most recent update was posted on 29 September 2025. 

## Figures and Tables

**Figure 1 healthcare-13-03062-f001:**
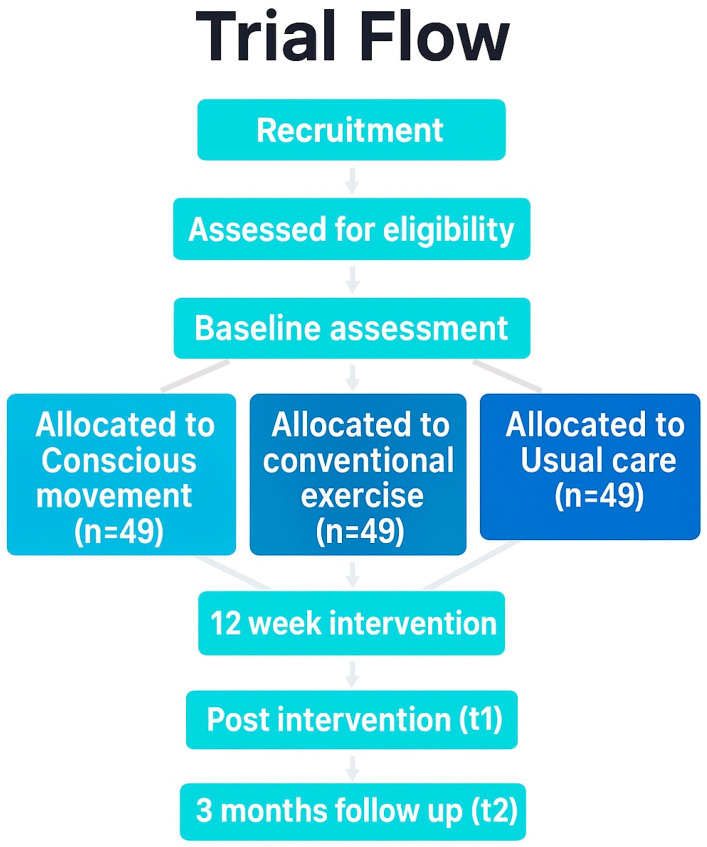
Flow diagram. Different background colors represent the different groups.

**Figure 2 healthcare-13-03062-f002:**
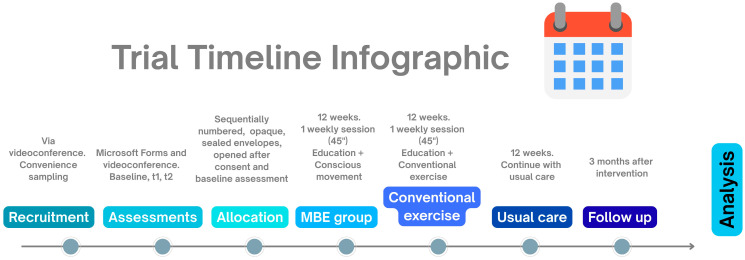
Trial timeline infographic. t1: end of 12-week program; t2: 3-month follow-up.

**Figure 3 healthcare-13-03062-f003:**
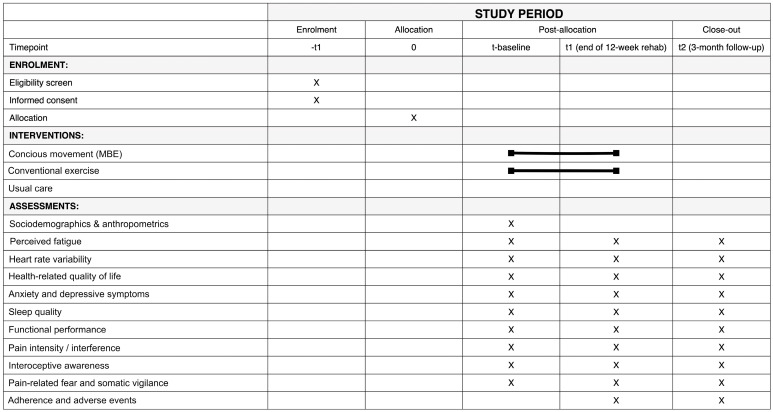
SPIRIT Figure showing the schedule of enrolment, interventions, and assessments. The black line indicates the duration of the intervention across the different time points.

## Data Availability

All study materials will be included as Appendix A with the article reporting the trial. The datasets generated and/or analyzed during the current study, as well as the data analysis code, will be made publicly available in the open repository of the University of Seville (https://idus.us.es/home, accessed on 12 October 2025). All shared datasets will be fully de-identified and will not contain any personally identifiable information.

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
