# Peer review of "Safety and Effectiveness of an Exercise-Based Telerehabilitation Program in Myalgic Encephalomyelitis and Post COVID Syndrome: Protocol for a Randomized Controlled Clinical Trial"

_healthcare, 2025, doi:10.3390/healthcare13233062_

Round 1

Reviewer 1 Report

Comments and Suggestions for Authors

Overall Comment

The authors have developed a highly detailed and meaningful study protocol. However, several issues require attention, and the text contains redundant expressions that need streamlining for better logic and conciseness.

  1. Title & Abstract

(1) The phrase "Related Variables" in the title is too vague. A more specific alternative is recommended.

(2) Stating the primary outcome as "fatigue" is problematic, as it is a concept. This should be rephrased throughout the manuscript to, for example, "the score on the Chalder Fatigue Scale (CFS) at the end of the 12-week intervention period."

(3) Please add the registration date and protocol version number after the trial registration number.

  1. Introduction

(1) The citation of "A Systematic Review of Trials... for Post-Acute COVID-19 Syndrome..." to support the statement "Low intensity aerobic exercise is widely recommended" for CFS/ME seems misplaced. While the similarity to Long COVID is understood, a more direct reference is needed here.

(2) Specifying potential Adverse Events (AEs) relevant to CFS/ME, supported by literature, would strengthen this section.

(3) While mindful movement-based exercise (MBE) is suggested to help fatigue and pain, providing quantitative data from existing studies would better support this claim.

(4) The primary aim states the MBE program will be evaluated for being "safer and more effective." However, the primary outcome is fatigue-related, with other metrics being secondary. This phrasing should be revised for consistency.

  1. Methods

The methodology is generally detailed but suffers from repetition and verbose descriptions that require rewriting.

(1) The statements cited in lines 102-104 must be specified with references.

(2) Please clarify the trial design: is it superiority, equivalence, or non-inferiority? Is it a single-center study?

(3) Eligibility Criteria:

1)How is a CFS/ME diagnosis confirmed? Is it based on prior medical diagnosis?

2)How is the disease duration for CFS/ME/PCS defined?

3) Withdrawal criteria are missing. Please refer to SPIRIT checklist item 15b.

(4) The section on Consent is repetitive and should be condensed.

(5) Sample Size Calculation: The description of the software is insufficient. Please provide the version number or a web link to the tool from the Clinical Epidemiology and Biostatistics Unit.

(6) How was the training duration for the three groups determined? Provide references. For the MBE group, how is adherence to "body awareness and relaxation" ensured or measured?

(7) Tele-Rehabilitation Adherence:

1)Beyond the measures mentioned, how is exercise quality or the achievement of target intensity monitored remotely? Clarify how "attendance to at least 70% of scheduled sessions" is defined and calculated.

2)Were participants provided with any tools for self-monitoring fatigue or discomfort?

(8) Outcome Measures:

1)The total time required to complete all assessments should be reported. Could lengthy testing affect participant performance or state?

2)Refine the description of resting HR and HRV measurement using the Welltory app (e.g., device placement, testing conditions).

3)Explain how the Modified Sit-and-Reach Test(distance measurement?) is administered reliably in a remote, online setting.

4)How to identify and handle invalid or unreasonable self-reported data collected online?

Comments on the Quality of English Language

The English could be improved to more clearly express the research.

Author Response

Reviewer 1

Overall Comment: The authors have developed a highly detailed and meaningful study protocol. However, several issues require attention, and the text contains redundant expressions that need streamlining for better logic and conciseness.

Thank you for the positive and constructive feedback. We have revised the manuscript to address the issues raised and to streamline the text by removing redundancies, tightening sentences, and clarifying the logical flow. All edits are marked in yellow and referenced in the point-by-point responses below.

Title & Abstract

Comment (1) The phrase "Related Variables" in the title is too vague. A more specific alternative is recommended.

Response: We agree. We have replaced this vague phrase with a more specific and concise title.

Changes in the manuscript:

Title Page (page #1) “Safety and Effectiveness of an Exercise-Based Telerehabilitation Program in Myalgic Encephalomyelitis and Post COVID Syndrome: Protocol for a Randomized Controlled Clinical Trial”

---

Comment (2): Stating the primary outcome as "fatigue" is problematic, as it is a concept. This should be rephrased throughout the manuscript to, for example, "the score on the Chalder Fatigue Scale (CFS) at the end of the 12-week intervention period."

Response: Thank you for raising this issue. As recommended, in the revised version of the manuscript, we define the primary outcome as the total score on the 14-item Chalder Fatigue Scale (CFQ-14) at 12 weeks (post-interventions).

Changes in the manuscript:

Abstract (page 1): The primary outcome is the total score on the 14-item Chalder Fatigue Scale at 12 weeks (post-intervention).

Methods, Sample size (page 5): Sample size estimation considered three study groups, a one-tailed test (a larger effect size is expected in the CMBE group), an α value of 0.05, a type II error (β) of 0.20 (80% power), and a precision of 6 (equivalent to a 15% change in the primary outcome, CFQ-14 total score at 12 weeks).

Methods, Outcomes (page 7): The primary outcome is the total score of the CFQ-14 (Spanish version) at 12 weeks (post-intervention).

Methods, Statistical methods (page 11): The primary outcome (CFQ-14 total score at 12 weeks, post-intervention) will be analyzed using linear mixed-effects models (LMMs) with fixed effects for group, time, and their interaction (group × time), and random intercepts to account for within-subject correlation.

----

Comment (3) Please add the registration date and protocol version number after the trial registration number.

Response: Thank you for the recommendation. As suggested, we have added this information to the abstract section.

Changes in the manuscript: Abstract - Trial registration: Registered at ClinicalTrials.gov on May 15, 2025(NCT06978582). Protocol version 4 (9/29/2025). Ethics Committee code: 2025-0180.

-----

Introduction

Comment (1): The citation of "A Systematic Review of Trials... for Post-Acute COVID-19 Syndrome..." to support the statement "Low intensity aerobic exercise is widely recommended" for CFS/ME seems misplaced. While the similarity to Long COVID is understood, a more direct reference is needed here.

Response: Thank you for observation. To remove any ambiguity and avoid overstatements, we revised the wording, separated the evidence referred to ME/CFS or PCS, and included a specific reference about the use of low-intensity exercise in individuals with ME/CFS.

Changes in the manuscript: (Introduction, page 2): Low-intensity exercise with strict pacing is recommended for individuals diagnosed with ME/CFS who are improving and wish to increase activity levels [13]. Such intervention may help relieve symptoms and prevent deconditioning in ME/CFS [11] and PCS [12].

-----

Comment (2) Specifying potential Adverse Events (AEs) relevant to CFS/ME, supported by literature, would strengthen this section.

Response: Thank you for this suggestion. We now specify adverse events relevant to ME/CFS and define post-exertional malaise (PEM) based on current guidance. The paragraph has been rewritten to improve readability.

Changes in the manuscript: (Introduction, page 2): However, PEM is common and greatly narrows the therapeutic window [14]. PEM involves a worsening of symptoms after minimal physical, cognitive, emotional, or social activity, with exacerbations typically occurring 12 to 48 hours later and lasting for days or weeks. Potential adverse events (AEs) therefore include PEM episodes and symptom flares, such as increased fatigue, cognitive difficulties, pain, sleep disturbance, and orthostatic intolerance. This requires careful dose titration, planned recovery, and continuous patient feedback [13].

Previous exercise trials in ME/CFS have reported AEs inconsistently and have often lacked patient-centered dosing approaches, undermining confidence in exercise-based rehabilitation [15–17]. Current clinical guidelines advise against generic graded exercise prescriptions and recommend caution, especially in severe disease [13].

----

Comment (3) While mindful movement-based exercise (MBE) is suggested to help fatigue and pain, providing quantitative data from existing studies would better support this claim.

Response: Thank you for the suggestion. We have provided brief quantitative context from previous research.

Changes in the manuscript: (Introduction, page 2,3): Evidence suggests that MBE produces similar effects on pain and fatigue to those of low-intensity aerobic training [29], although direct head-to-head comparisons with active exercise controls remain scarce [29,30]. Meta-analyses have shown small-to-moderate reductions in fatigue with MBE in ME/CFS and PCS (SMD −0.44, 95% CI −0.63 to −0.25) [29]. Beyond physical impact, MBE incorporates benefits associated with meditation, mindfulness practices, and conscious breathing.

---

Comment (4) The primary aim states the MBE program will be evaluated for being "safer and more effective." However, the primary outcome is fatigue-related, with other metrics being secondary. This phrasing should be revised for consistency.

Response: Thank you for this comment. We totally agree. Accordingly, we have revised the study aims to align with the primary outcome. Secondary aims are also included.

Changes in the manuscript: (Introduction, page 3): The primary aim is to determine whether a MBE telerehabilitation program reduces fa-tigue severity, as measured with the 14-item Chalder Fatigue Questionnaire (CFQ-14), more effectively than conventional low-intensity exercise after 12 weeks of intervention. Secondary aims are to: (i) describe safety outcomes, with a focus on AEs under a pacing and recovery strategy; (ii) assess changes in dysautonomia-related symptoms; and (iii) examine the durability of effects 3 months after intervention. We hypothesize that the MBE program will enhance autonomic balance and interoceptive awareness compared with conventional exercise, facilitating more adaptative pacing and recovery behaviors, and therefore reducing the frequency and severity of PEM. Consequently, greater im-provements in CFQ-14 scores are expected in the MBE group at 12 weeks. Although low AEs rates are anticipated under careful pacing and monitoring across groups, we hy-pothesize that the durability of effects will depend on participants’ adherence to the program after completion of the intervention.

----

Methods

The methodology is generally detailed but suffers from repetition and verbose descriptions that require rewriting.

Comment (1) The statements cited in lines 102-104 must be specified with references.

Response: Thank you for the insight. Following your recommendation, we have now specified the proper references.

Changes in the manuscript: (Methods, Trial design, page 3): The protocol adheres to the Standard Protocol Items Recommendations for Interventional Trials (SPIRIT) guideline [35], the Template for Intervention Description and Replication (TIDieR) checklist [36], and the Consensus on Exercise Reporting Template (CERT) [37].

----

Comment (2) Please clarify the trial design: is it superiority, equivalence, or non-inferiority? Is it a single-center study?

Response: Thank you for the request. This has been now clarified in the abstract and method sections of the revised manuscript

Changes in the manuscript: Abstract - This is a prospective, single-blind, three-arm, parallel, superiority randomized clinical trial.

Section 2.1 Trial design - A single-blind, three-arm, parallel-group, superiority randomized controlled trial will be conducted.

Section 2.2 Study setting - The study is decentralized and fully remote. A single coordinating center will oversee recruitment, randomization, intervention delivery, data management, and safety mon-itoring. No in-person visits will be required.

-----

Comment (3) Eligibility Criteria:

- How is a CFS/ME diagnosis confirmed? Is it based on prior medical diagnosis?

Response: Thank you for the question. A new section has been included in the methods under the subheading “Diagnosis confirmation and disease duration.”

Changes in the manuscript: (Methods, page 5): 2.4 Diagnosis confirmation and disease duration. A prior formal ME/CFS or PCS diagnosis is not required due to very limited specialist access withing European clinical settings [39,40]. All participants must provide documentation of a prior medical evaluation reasonably excluding alternative diagnoses (e.g., post-infectious organ damage, cardiovascular or neurological disease, cancer). The research team will not perform medical diagnoses. The lead investigator will verify that participants’ medical history meets the diagnostic criteria using a structured checklist and review of available records.

- How is the disease duration for CFS/ME/PCS defined?

Response: Thank you for your comment. A clear description of how the disease duration for ME/CFS and PCS was defined has now been added to the methods section.

Changes in the manuscript: Disease duration will be defined as the number of months from the self-reported onset of persistent symptoms meeting diagnostic criteria to the baseline assessment. For PCS, the date of the acute SARS-CoV-2 infection will also be recorded.

----

Comment 3) Withdrawal criteria are missing. Please refer to SPIRIT checklist item 15b.

Response: To properly address this issue, we have expanded Section 2.10.4 (Criteria for discontinuing or modifying allocated interventions). To avoid excessive information in the main text and to comply with the journal’s word limit requirements, some of the criteria for discontinuing or modifying the allocated intervention, the decision process, and post-withdrawal assessments, are described in greater detail in S4 File.

Changes in the manuscript: (Methods, Criteria for discontinuing or modifying allocated interventions, page 7):Participants may pause or modify sessions at any time according to pacing principles. Therapists may recommend temporary pauses or adjustments based on symptoms. No cross-over between study arms will be permitted. The principal investigator (PI) will decide on discontinuation if safety concerns arise. Withdrawals will be respected without requiring justification. Temporary interruptions due to intercurrent illness or symptom flare will be permitted, with resumption when appropriate. Enrollment in other clinical trials will not be allowed. Any relevant changes in medical treatment will be documented but will not necessarily lead to withdrawal unless recommended by a physician. In the event of AEs, participants will be offered an individual videocall to address causes and adjust the intervention accordingly. More specific details are included in the S4. File.

S4 File: Triggers to modify the intervention, pause sessions or discontinue the allocated intervention

Triggers to modify the intervention (dose titration, switching chair/standing options, longer rests, lower frequency or duration):

  • Suffering an increase over 2/10 or reaching 7/10 in the Numeric Rating Scale for Fatigue or Pain related to sessions.
  • Other non-serious adverse events judged related to sessions.
  • Transient mild flare triggered by non-study related causes.
  • Participant-reported excessive burden or two consecutive sessions aborted due to symptoms.

Process after AEs or safety concerns. We will offer a one-to-one safety videocall to review symptoms and adjust the plan

Triggers to temporarily pause sessions:

  • Intercurrent medical condition not related to the trial requiring short-term rest.

Triggers to discontinue the allocated intervention:

  • Suffering an increase over 5/10 or reaching 9/10 in the Numeric Rating Scale for Fatigue or Pain related to sessions.
  • Other serious adverse events or clinically significant deterioration judged related to the intervention.
  • New medical diagnosis or condition that contraindicates exercise (per treating physician).
  • Participant request to stop the program.
  • Start of another structured rehabilitation program or treatment that conflicts with the trial.

----

Comment (4) The section on Consent is repetitive and should be condensed.

Response: Thank you for the suggestion. As suggested in this specific comment, as well as more generally for the entire Methods section, the manuscript has been thoroughly revised to eliminate redundancies and thereby enhance clarity and readability.

Changes in the manuscript: Consent-related passages were eliminated. Please, see also the revised manuscript for changes within the entire paper.

----

Comment (5) Sample Size Calculation: The description of the software is insufficient. Please provide the version number or a web link to the tool from the Clinical Epidemiology and Biostatistics Unit.

Response: Software information has been updated. A new reference has been added with a direct web link to the Clinical Epidemiology and Biostatistics Unit tool.

Changes in the manuscript: (Methods, Sample size, page 5):  Calculations were made using software developed by the Clinical Epidemiology and Biostatistics Unit of the Complexo Hospitalario Universitario de A Coruña, Spain [42].

Reference [42] López-Calviño, B.; Pita-Fernández, S.; Pértega Díaz, S.; Seoane-Pillado, T. Calculadora de Tamaño Muestral. Unidad de Epidemiología Clínica y Biostatística, Complexo Hospitalario Universitario de A Coruña. https://www.google.com/url?sa=t&source=web&rct=j&opi=89978449&url=https://www.fisterra.com/mbe/investiga/9muestras/tamano_muestral.xls&ved=2ahUKEwiHic_RxtQAxXCA9sEHehUJD8QFnoECB4QAQ&usg=AOvVaw3jJbvg9TNKhAo4FHkboLZ1

---

(6) How was the training duration for the three groups determined? Provide references. For the MBE group, how is adherence to "body awareness and relaxation" ensured or measured?

Response: Thank you for your comment. In line with previous research in ME/CFS and PCS, the intervention duration was set at 12 weeks, a period considered sufficient for participants to develop pacing strategies and interoceptive skills. Adherence to "body awareness and relaxation" represents a subjective experience and is inherently challenging to quantify; however, we will promote these practices throughout the intervention using structured guidance, home-practice exercises, and session-based reinforcement. Details regarding intervention duration and adherence strategies have been added to relevant sections of the manuscript.

Changes in the manuscript: (Methods, Interventions, page 6): All sessions will be delivered live via Microsoft Teams, with groups of ≤ 7–8 participants, once weekly for 12 consecutive weeks, consistent with previous research in ME/CFS and PCS [29,43].

(Methods, Interventions, page 6): To enhance proprioceptive and interoceptive awareness, therapists will follow a brief checklist (posture, breath, body-scan, pacing, and cognitive reappraisal cues) at the be-ginning of each session. Body awareness will be reinforced through short post-session exercises (attention to bodily sensations; effort rating using sensations; perceived calm), and weekly home-practice using prerecorded materials.

(Methods, Other secondary outcomes, page 9): Interoceptive awareness and changes in body awareness will be measured using the Multidimensional Assessment of Interoceptive Awareness (MAIA) scale.

----

Comment (7) Tele-Rehabilitation Adherence:

- Beyond the measures mentioned, how is exercise quality or the achievement of target intensity monitored remotely? Clarify how "attendance to at least 70% of scheduled sessions" is defined and calculated.

Response: Exercise quality is monitored in real time by the therapist during live group sessions (≤7–8 participants), allowing visual supervision and immediate corrective feedback. Exercise intensity is individualized and symptom-dependent, following pacing principles, rather than a prescribed fixed target. Therapists will guide participants using in-session ratings of perceived exertion RPE (2-3/10) and simple talk-test cues to maintain a safe and comfortable while minimizing the risk of symptom exacerbation. Information on how attendance will be calculated has been added

Changes in the manuscript: (Methods, Interventions, page 6): All sessions will be delivered live via Microsoft Teams, with groups of ≤ 7–8 participants, once weekly for 12 consecutive weeks, consistent with previous research in ME/CFS and PCS [29,43].

(Methods, Interventions, page 6) Attendance to ≥70% of the 12 scheduled sessions will be considered adequate exposure. A session will be considered attended if the participant joins the live class and takes part at least partially (≥15 min or completion of ≥1 exercise block), in line with pacing allowances. Attendance will be tracked via Microsoft Teams and verified by therapist logs.

S4. File - Symptom-contingent dosing: There is not a fixed intensity target. Adjust it depending on actual energy levels and keep intensity very light to light (2/3 Borg scale); use the talk test (able to speak in full sentences).

----

  • Were participants provided with any tools for self-monitoring fatigue or discomfort?

Response: Thank you for your insight here. This aspect has been now clarified in the revised version of the manuscript. Please, see changes made below.

Changes in the manuscript: (Methods, page 7): The Welltory app will be used to log daily fatigue, discomfort, and PEM warnings. Entries are time-stamped and symptom logs are aligned with HRV data for safety reviews.

----

Comment (8) Outcome Measures:

  • The total time required to complete all assessments should be reported. Could lengthy testing affect participant performance or state?

Response: Thank you for inviting us to expand on this section. Information has been added in different sections of the manuscript to clarify this aspect.

Changes in the manuscript: (Methods, Outcomes, page 7): Data will be collected using Microsoft Forms and videoconference assessments. Participants will complete online questionnaires within a 2-week window at their own pace. The PI will monitor submissions for completeness, and plausibility/inconsistency issues. Implausible or inconsistent entries prompt participant queries. Corrections will be documented without overwriting original data. During this 2-week period, participants will attend a ~20-minute videocall to assess HRV first and then functional performance in a standardized sequence: Modified Sit-and-Reach Test (MSRT), One-Leg Stand Test (OLST), and 60-second Sit-to-Stand Test (STS-60), to limit fatigue carry-over. Videocalls will be scheduled Monday–Thursday, 10:00–13:00 and 16:00–19:00 avoiding meals and sleep times. Rescheduling will be allowed during flares. This approach minimizes participant burden and reduces the risk that lengthy procedures may influence performance or clinical status.

(Methods, Outcomes, page 9): Lower-limb strength and endurance will be evaluated with the STS-60. Participants will perform repeated sit-to-stand movements from a standard armless chair for 60 seconds, with arms crossed over the chest. Effort will be self-regulated to a perceived exertion of approximately 6/10 (moderate). Participants will be instructed to maintain a comfortable pace and may slow down or pause if symptoms increase. The total number of valid repetitions (full stand and controlled sit) will be recorded.

  • Refine the description of resting HR and HRV measurement using the Welltory app (e.g., device placement, testing conditions).

Response: As suggested, the description of resting HR and HRV measurements has been refined.

Changes in the manuscript: (Methods, Outcomes, page 8): Resting heart rate (HR) and HRV will be assessed using the Welltory mobile app (V4.16.100, Welltory Inc. Redwood City, California, EE. UU). Resting HR values will be obtained via photoplethysmography through the smartphone camera, while the time-domain HRV indices will be derived as proxies of autonomic balance. Previous studies report excellent agreement to estimate resting HRV using this method when compared with established referenced tools, including electrocardiography and medical-grade HR monitors (intraclass correlation coefficient > 0.90) [46]. Measurements will be made at the start of the videocall. Participants will be instructed to sit quietly, breathe spontaneously, rest their forearms on their thighs, and hold the smartphone in their dominant hand, placing the index finger on the camera as indicated by the app. Speaking or moving during the recording will not be permitted. The 300-beat setting will be used to maximize precision. Both resting HR and HRV will be extracted from the same measurement.

  • Explain how the Modified Sit-and-Reach Test (distance measurement?) is administered reliably in a remote, online setting.

Response: Thank you for your input. Remote assessment of functional performance has been shown to be both feasible and reliable when delivered under standardized protocols and real-time supervision description. Specific information on how the Modified Sit-and-Reach Test will be administered is now included.

Changes in the manuscript: (Methods, Outcomes, page 9): Remote assessment of functional performance has been shown to be both feasible and reliable when delivered under standardized protocols and real-time supervision [50]. The study will follow international recommendations for remote physical performance testing [50]. Before assessment, participants will receive clear instructions to prepare a well-lit space and position the camera to provide a full-body lateral view. Each test will be demonstrated in advance, and corrective feedback will be provided as needed to ensure proper execution.

Posterior chain flexibility will be measured using the MSRT. Participants will sit on a firm surface (e.g., floor) with legs extended, ankles neutral, and a one-fist gap between the knees. A small object (≤3 –4 cm diameter) will be placed between the legs near the pelvis, aligned with the midline. With arms extended, participants will slide the object forward along the midline as far as possible while keeping knees extended and avoiding bouncing. The assessor will view from the side to ensure correct execution. Using a tape measure, the distance will be recorded in centimeters from the pubic symphysis (over clothing) to the nearest edge of the object at the farthest reach. Testing will stop immediately if pain or cramping occurs. Higher values indicate greater flexibility.

---

  • How to identify and handle invalid or unreasonable self-reported data collected online?

Response: Thank you for your comment. As recommended, the description of this aspect has been clarified in the revised version of the paper.

Changes in the manuscript: (Methods, Outcomes, page 7): The PI will monitor submissions for completeness, and plausibility/inconsistency issues. Implausible or inconsistent entries prompt participant queries. Corrections will be documented without overwriting original data.

We hope to have properly clarified all the reviewers comments and concerns.

Reviewer 2 Report

Comments and Suggestions for Authors

Abstract

The abstract could be further improved by slightly shortening the background section and emphasizing more clearly the study’s innovation and expected contribution to clinical practice. Consider including one or two sentences that specify how the proposed tele-rehabilitation program differs from other existing protocols in terms of structure or theoretical basis. Also, make sure that abbreviations are used consistently and defined at first mention (e.g., MBE, PCS).

Introduction

The introduction effectively presents the clinical overlap between CFS/ME and PCS, as well as the rationale for integrating mindful movement and tele-rehabilitation. However, it would benefit from a more concise articulation of the research gap, what remains unknown despite the existing studies on tele-rehabilitation and mind–body exercise. In addition, the section could be strengthened by linking the conceptual framework (interoception, pacing, dysautonomia) more directly to the study hypothesis, highlighting how these mechanisms may influence the expected outcomes.

Methods

The description of the randomization and blinding procedures, while comprehensive, could be simplified to improve readability. Currently, these processes are reiterated across multiple subsections, which slightly disrupts the flow of information. Condensing this content into a single, clearly articulated paragraph would make the protocol more concise without compromising detail.

In addition, the manuscript would benefit from more explicit specification of the criteria used to discontinue or modify the intervention. Defining concrete thresholds or clinical indicators, such as symptom exacerbation levels, fatigue intensity scores, or heart rate variability parameters, would improve the replicability and clinical transparency of the study. Clear operationalization of these criteria is particularly important for research involving vulnerable populations such as individuals with CFS/ME and PCS, where the balance between exertion and safety is delicate.

It would also be useful to clarify whether the research team plans to perform preliminary reliability or calibration checks for the online functional assessments, such as the Modified Sit-and-Reach Test (MSRT), One-Leg Stand Test (OLST), and Sit-to-Stand Test (STS-60). Since these measurements will be conducted remotely, variability in participant setup, camera positioning, or environmental factors could influence outcomes. A brief statement describing standardization strategies, inter-rater checks, or pilot testing would strengthen the methodological robustness of the trial.

Finally, while the data management plan is commendably thorough and compliant with data protection standards, it could be expanded to specify the expected duration of data storage and the timeframe during which de-identified datasets will remain accessible after publication. This addition would reinforce the study’s commitment to transparency, reproducibility, and long-term research integrity, aligning it with current open science practices.

Discussion

The discussion could be refined by expanding on how this trial’s findings may contribute to the development of future clinical guidelines or inform public health strategies for CFS/ME and PCS. Additionally, the discussion might briefly address how the results could be compared or integrated with previous similar interventions (e.g., other telehealth-based or mindfulness-based trials). Finally, a short paragraph emphasizing how this protocol aligns with the broader digital health transformation in rehabilitation would enhance its applied value.

Limitations
In the limitations section, consider discussing potential selection bias, as participation requires digital literacy and reliable internet access, and how this might affect generalizability. Also, mention the potential for self-report bias due to online assessments, and outline how these effects will be minimized (e.g., through standardized instructions, training sessions, or triangulation with objective data).

Conclusion
To make it more impactful, consider adding one forward-looking statement emphasizing how the protocol may set a foundation for future multi-center or cross-national trials. The final sentence could better underscore the potential health policy implications, especially regarding the integration of tele-rehabilitation as a standard option for patients with chronic fatigue-related disorders.

Author Response

Reviewer 2

Comments and Suggestions for Authors

First, we would like to thank the reviewer for the constructive feedback. We have revised the paper to address all issues. Changes in the revised version of the manuscript are marked in yellow and listed in this point-by-point responses document.

Comment (1) Abstract: The abstract could be further improved by slightly shortening the background section and emphasizing more clearly the study’s innovation and expected contribution to clinical practice. Consider including one or two sentences that specify how the proposed tele-rehabilitation program differs from other existing protocols in terms of structure or theoretical basis. Also, make sure that abbreviations are used consistently and defined at first mention (e.g., MBE, PCS).

Response: Given the word count limitations in the abstract, as suggested, we have focused on highlighting the need for novel therapeutic approaches for ME/CFS and PCS, emphasizing the role of mindful movement in exercise-based approaches. This aspect has been underscored in both the introduction and discussion sections of the abstract.”

Changes in the manuscript: Please, see the revised version of the abstract (page 1)

----

Comment (2) Introduction: The introduction effectively presents the clinical overlap between CFS/ME and PCS, as well as the rationale for integrating mindful movement and tele-rehabilitation. However, it would benefit from a more concise articulation of the research gap, what remains unknown despite the existing studies on tele-rehabilitation and mind–body exercise. In addition, the section could be strengthened by linking the conceptual framework (interoception, pacing, dysautonomia) more directly to the study hypothesis, highlighting how these mechanisms may influence the expected outcomes.

Response: Thank you again for your recommendations. We have modified the introduction section following your suggestion to strengthen this section and established a clear link between the conceptual framework to the study hypothesis and aims.

Changes in the manuscript: (Introduction, page 3): Despite promising findings, key evidence gaps remain for the efficacy of telerehabilitation and MBE. Telerehabilitation is under characterized in ME/CFS, and most existing evidence derives from post-COVID cohorts. Few studies have compared MBE with active exercise interventions. Reporting of safety and AEs is inconsistent, and mechanisms such as interoceptive awareness and autonomic balance are rarely assessed. Similarly, the durability of effects beyond the treatment period is still uncertain.

To address these gaps, this protocol proposes a three-arm, head-to-head design, with PEM-aware dosing, strict AEs monitoring, inclusion of process measures (interoceptive awareness, heart rate variability -HRV), and follow-up at 3 months. The primary aim is to determine whether a MBE telerehabilitation program reduces fatigue severity, as measured with the 14-item Chalder Fatigue Questionnaire (CFQ-14), more effectively than conventional low-intensity exercise after 12 weeks of intervention. Secondary aims are to: (i) describe safety outcomes, with a focus on AEs under a pacing and recovery strategy; (ii) assess changes in dysautonomia-related symptoms; and (iii) examine the durability of effects 3 months after intervention. We hypothesize that the MBE program will enhance autonomic balance and interoceptive awareness compared with conventional exercise, facilitating more adaptative pacing and recovery behaviors, and therefore reducing the frequency and severity of PEM. Consequently, greater improvements in CFQ-14 scores are expected in the MBE group at 12 weeks. Although low AEs rates are anticipated under careful pacing and monitoring across groups, we hypothesize that the durability of effects will depend on participants’ adherence to the program after completion of the intervention.

----

Comment (3) Methods: The description of the randomization and blinding procedures, while comprehensive, could be simplified to improve readability. Currently, these processes are reiterated across multiple subsections, which slightly disrupts the flow of information. Condensing this content into a single, clearly articulated paragraph would make the protocol more concise without compromising detail. In addition, the manuscript would benefit from more explicit specification of the criteria used to discontinue or modify the intervention. Defining concrete thresholds or clinical indicators, such as symptom exacerbation levels, fatigue intensity scores, or heart rate variability parameters, would improve the replicability and clinical transparency of the study. Clear operationalization of these criteria is particularly important for research involving vulnerable populations such as individuals with CFS/ME and PCS, where the balance between exertion and safety is delicate. It would also be useful to clarify whether the research team plans to perform preliminary reliability or calibration checks for the online functional assessments, such as the Modified Sit-and-Reach Test (MSRT), One-Leg Stand Test (OLST), and Sit-to-Stand Test (STS-60). Since these measurements will be conducted remotely, variability in participant setup, camera positioning, or environmental factors could influence outcomes. A brief statement describing standardization strategies, inter-rater checks, or pilot testing would strengthen the methodological robustness of the trial. Finally, while the data management plan is commendably thorough and compliant with data protection standards, it could be expanded to specify the expected duration of data storage and the timeframe during which de-identified datasets will remain accessible after publication. This addition would reinforce the study’s commitment to transparency, reproducibility, and long-term research integrity, aligning it with current open science practices.

Response: Thank you for your helpful suggestions for this section. Following your comment and those of another reviewer, the description of the randomization and blinding procedures has been simplified to avoid redundancies and improve the overall readability of this section. Similarly, the section devoted to explaining the specific thresholds and indicators to modify or discontinue the intervention has been developed. To avoid excessive information in the main text and to comply with the journal’s word limit requirements, some of the criteria for discontinuing or modifying the allocated intervention, the decision process, and post-withdrawal assessments, are described in greater detail in the S4 File.

To ensure consistency in both evaluations and intervention delivery, preparatory training sessions will be conducted before recruitment. Additional details on the assessment procedures have been added for clarification. Thank you for noting the data management plan; it has been updated to align with current open science practices.”

Changes in the manuscript:

S4. File - Triggers to modify the intervention, pause sessions or discontinue the allocated intervention

Triggers to modify the intervention (dose titration, switching chair/standing options, longer rests, lower frequency or duration):

  • Suffering an increase over 2/10 or reaching 7/10 in the Numeric Rating Scale for Fatigue or Pain related to sessions.
  • Other non-serious adverse events judged related to sessions.
  • Transient mild flare triggered by non-study related causes.
  • Participant-reported excessive burden or two consecutive sessions aborted due to symptoms.

Process after AEs or safety concerns. We will offer a one-to-one safety videocall to review symptoms and adjust the plan

Triggers to temporarily pause sessions:

  • Intercurrent medical condition not related to the trial requiring short-term rest.

Triggers to discontinue the allocated intervention:

  • Suffering an increase over 5/10 or reaching 9/10 in the Numeric Rating Scale for Fatigue or Pain related to sessions.
  • Other serious adverse events or clinically significant deterioration judged related to the intervention.
  • New medical diagnosis or condition that contraindicates exercise (per treating physician).
  • Participant request to stop the program.
  • Start of another structured rehabilitation program or treatment that conflicts with the trial.

(Methods, page 7): Data will be collected using Microsoft Forms and videoconference assessments. Participants will complete online questionnaires within a 2-week window at their own pace. The PI will monitor submissions for completeness, and plausibility/inconsistency issues. Implausible or inconsistent entries prompt participant queries. Corrections will be documented without overwriting original data. During this 2-week period, participants will attend a ~20-minute videocall to assess HRV first and then functional performance in a standardized sequence: Modified Sit-and-Reach Test (MSRT), One-Leg Stand Test (OLST), and 60-second Sit-to-Stand Test (STS-60), to limit fatigue carry-over. Videocalls will be scheduled Monday–Thursday, 10:00–13:00 and 16:00–19:00 avoiding meals and sleep times. Rescheduling will be allowed during flares. This approach minimizes participant burden and reduces the risk that lengthy procedures may influence performance or clinical status.

(Methods, page 8): Remote assessment of functional performance has been shown to be both feasible and reliable when delivered under standardized protocols and real-time supervision [50]. The study will follow international recommendations for remote physical performance testing [50]. Before assessment, participants will receive clear instructions to prepare a well-lit space and position the camera to provide a full-body lateral view. Each test will be demonstrated in advance, and corrective feedback will be provided as needed to ensure proper execution.

(Methods, page 12): The full study protocol and data collection materials will be made publicly available to promote transparency and reproducibility. De-identified datasets and analysis code will be deposited in the institutional open-access repository (https://idus.us.es/home) under a persistent identifier and will remain accessible without time restriction. Source data containing personal information will be securely stored on university servers for 3 years after study close-out and will then be deleted or irreversibly anonymized.

---

Comment (4) Discussion: The discussion could be refined by expanding on how this trial’s findings may contribute to the development of future clinical guidelines or inform public health strategies for CFS/ME and PCS. Additionally, the discussion might briefly address how the results could be compared or integrated with previous similar interventions (e.g., other telehealth-based or mindfulness-based trials). Finally, a short paragraph emphasizing how this protocol aligns with the broader digital health transformation in rehabilitation would enhance its applied value.

Response: As suggested, the discussion section has been refined expanding on the possible contributions of the trial findings on the emerging body of evidence in this field.

Changes in the manuscript: (Discussion, Page 13): In line emerging research directions [67,68], this study proposes a symptom-titrated telerehabilitation approach that dynamically adjusts exercise load through clear, structured, and comprehensive pacing strategies tailored to individual’s clinical response. The MBE program has been designed to address key limitations of previous telerehabilitation programs and provide a truly personalized care model for patients with ME/CFS and PCS. Therefore, the trial findings may help shape future clinical guidelines and inform public health strategies into standard management of CFS/ME and PCS.

----

Comment (5) Limitations: In the limitations section, consider discussing potential selection bias, as participation requires digital literacy and reliable internet access, and how this might affect generalizability. Also, mention the potential for self-report bias due to online assessments, and outline how these effects will be minimized (e.g., through standardized instructions, training sessions, or triangulation with objective data).

Response: Thank you for pointing out these important aspects. The study limitations paragraph has been updated with new information.

Changes in the manuscript: (Discussion, Page 13): Nevertheless, this approach presents inherent limitations. In line with other telemedicine trials, participation requires stable internet access and basic digital literacy, which may restrict the generalizability of findings [69]. Moreover, individuals with cognitive impairments may face challenges engaging with online content, scheduling videoconferences, or using digital applications [50]. These issues may be further complicatedwhen digital tools such as Welltory, are not available in the participant’s native language. Online assessments may also limit the precision of certain outcome measures compared with face-to-face evaluations. Although all instruments in this trial have been validated, some may not achieve the same accuracy as gold-standard methods available in onsite settings. Biological sampling is also precluded in an online clinical trial. Self-report outcomes are susceptible to expectation and recall bias; we aim to mitigate through triangulation with objective and observer-rated measures. The absence of in-person group dynamics may further reduce potential psychosocial benefits derived from peer interaction and shared experiences, which are commonly observed in traditional rehabilitation environments [70]. These limitations should be carefully considered when interpreting trial results and planning future iterations of the interventions.

---

Comment (6) Conclusion: To make it more impactful, consider adding one forward-looking statement emphasizing how the protocol may set a foundation for future multi-center or cross-national trials. The final sentence could better underscore the potential health policy implications, especially regarding the integration of tele-rehabilitation as a standard option for patients with chronic fatigue-related disorders.

Response: As suggested, a brief statement has been included as a conclusion section.

Changes in the manuscript:

Conclusion (page 14) - This trial aims to examine the effects of a 12-week mindful exercise telerehabilitation program compared to conventional exercise and usual care in individuals with ME/CFS and PCS. Beyond testing feasibility and clinical outcomes, the study seeks to elucidate whether an interoceptive, symptom-titrated approach can support safer and more sustainable physical activity engagement.

Findings from this trial may help refine personalized rehabilitation strategies and inform the integration of telerehabilitation into routine clinical pathways for ME/CFS and PCS. By providing a structured, evidence-based model of remote delivery, the protocol could serve as a foundation for future multicenter or cross-national collaborations and guide health policy decisions toward broader access to individualized, home-based rehabilitation options.

We hope to have properly clarified all the reviewers comments and concerns.
